# Do the Causes of Spontaneous Preterm Delivery Affect Placental Inflammatory Pathology and Neonatal Outcomes?

**DOI:** 10.3390/diagnostics12092126

**Published:** 2022-09-01

**Authors:** Il-Yeo Jang, Hye-Ji Jung, Ji-Hee Sung, Suk-Joo Choi, Soo-Young Oh, Jung-Sun Kim, Cheong-Rae Roh

**Affiliations:** 1Department of Obstetrics and Gynecology, Samsung Medical Center, Sungkyunkwan University School of Medicine, Seoul 06351, Korea; 2Department of Pathology, Samsung Medical Center, Sungkyunkwan University School of Medicine, Seoul 06351, Korea

**Keywords:** preterm delivery, preterm labor, preterm premature rupture of membranes, incompetent internal os of the cervix, placental pathology, neonatal outcomes

## Abstract

Objective: To investigate the severity of histologic chorioamnionitis /funisitis according to the indication for preterm delivery and their corresponding neonatal outcomes. Method: This study included 411 singleton women who delivered between 21^+0^ and 31^+6^ week of gestation due to preterm labor (PTL, *n* = 165), preterm premature rupture of membranes (PPROM, *n* = 202), or incompetent internal os of the cervix (IIOC, *n* = 44). The primary outcome measure was the rate of severe histological chorioamnionitis/funisitis. Secondary outcome measure was neonatal outcomes including neonatal and infant death, and neonatal composite morbidity. Results: The PPROM group demonstrated a higher rate of severe histological chorioamnionitis/funisitis compared to the PTL group (severe histological chorioamnionitis; PPROM, 66.3% vs. PTL, 49.1%, *p* = 0.001, severe funisitis; PPROM, 44.1% vs. PTL, 23.6%, *p* < 0.001) and this remained significant after multivariable analysis (severe histologic chorioamnionitis, OR 2.367, 95% CI 1.517–3.693; severe funisitis, OR 2.668, 95% CI 1.684–4.226). For neonatal outcomes only, a higher rate of patent ductus arteriosus was observed in the IIOC group compared to the PTL and PPROM groups (IIOC, 77.3% vs. PTL, 54.0% vs. PPROM, 54.0%, *p* = 0.043) and this remained significant after multivariable analysis. Conclusion: Indication of spontaneous preterm delivery might affect the placental inflammatory pathology and neonatal morbidity.

## 1. Introduction

Preterm delivery is defined as delivery before 37 weeks of gestation, and its incidence is gradually increasing. Despite improvements in neonatal care, preterm birth remains the leading cause of neonatal morbidity and mortality [1,2]. More than two-thirds of preterm births are due to spontaneous preterm delivery, and up to 80% of them are associated with preterm labor (PTL), preterm premature rupture of membranes (PPROM), and incompetent internal os of the cervix (IIOC). PTL is the most common cause of spontaneous preterm delivery and is defined as regular uterine contractions with cervical change before 37 weeks of gestation. PPROM is defined as membrane rupture before 37 weeks of gestation and causes 30~35% of preterm delivery [3]. IIOC is painless cervical dilatation without PTL or PPROM and is also an important cause of preterm delivery. There are multiple risk factors for spontaneous preterm delivery, among which intrauterine infections and inflammation are well-known risk factors. It is reported that particular cytokines which can weaken the placental barrier and fetal membranes are released when chorioamnionitis or funisitis develops, and this might cause the premature rupture of membranes [4,5]. Previous studies have demonstrated that intrauterine infections can cause placental inflammation by affecting the amnion, chorio-decidua, and extra-umbilical cord in the placenta [6,7,8,9,10]. For this reason, histologic chorioamnionitis and funisitis are histological hallmarks of maternal and fetal inflammatory responses, respectively. A study by Kacerovsky et al. showed that severe intra-amniotic infection was related to severe placental inflammation in women with PPROM [11]. As incidence of intra-amniotic infection/inflammation is different in PTL, PPROM, and IIOC [12,13,14], the incidence and severity of placental inflammation might differ among these three groups. A study compared preterm gestations that delivered before 34 weeks of gestation due to PTL or PPROM, and they demonstrated a higher rate of acute histologic chorioamnionitis and funisitis in patients with PPROM than in those with PTL [15]. They also revealed a significantly higher rate of intra-amniotic infection in women with PPROM than in those with PTL. However, only a few studies have compared placental pathology among women with PTL, PPROM, and IIOC.

As the placenta plays an important role as a fetal organ in the uterus, placental inflammation may affect the normal development and maturation of the fetus. Some studies have attempted to determine the association between pathological changes in the placenta and neonatal outcomes and have shown conflicting results. A study by Bersani et al. demonstrated that placental inflammation may increase the risk of perinatal morbidity and mortality [16]. In contrast, Lee et al. demonstrated an increasing incidence of neonatal morbidities with increasing stages of histological chorioamnionitis and funisitis in women with PPROM, but this was not significant after adjusting for gestational age at delivery [17]. Although several studies have been conducted on the relationship between the severity of placental pathology and neonatal prognosis in preterm mothers, there are few studies on how the severity of placental pathology differs depending on the cause of preterm delivery.

In this study, we hypothesized that the severity of histologic chorioamnionitis/funisitis and neonatal outcomes might be different between women who delivered due to PTL, PPROM, and IIOC. We compared the rate of histologic chorioamnionitis or funisitis and neonatal outcomes among women with PTL, PPROM, and IIOC.

## 2. Materials and Methods

This retrospective cohort study was conducted at a tertiary referral hospital in Seoul, Korea, from January 2012 to December 2018. Consecutive pregnant women with singleton gestations who delivered between 21 weeks and 0day and 31 weeks and 6 days were considered eligible and screened by reviewing electronic medical records. This study was approved by the Institutional Review Board for Clinical Research at Samsung medical center (IRB No 2022-04-146) and exemption for informed consent was granted because this is a retrospective study. Gestational age was determined by the patient’s last menstrual period or crown-rump length on ultrasonography in the first trimester. Among them, women who delivered due to maternal-fetal indications and those with fetuses who had major anomalies were excluded.

We divided the study population into three groups according to the indication for preterm delivery: preterm labor (PTL), preterm premature rupture of membranes (PPROM), and incompetent internal os of the cervix (IIOC).

The indication for preterm delivery was the diagnosis made at the time of the initial presentation of threatened preterm delivery. PTL was defined as regular uterine contractions more than five times in an hour with cervical change, and PPROM was defined as positive on nitrazine or amnisure test, or leaking or pooling of amniotic fluid from the cervical os. IIOC was defined as painless cervical dilatation without PTL or PPROM.

Maternal characteristics, including age, parity, type of conception (spontaneous or assisted reproduction techniques), body mass index (BMI) before pregnancy, use of antenatal corticosteroids (ACS), gestational diabetes mellitus, and hypertensive disorders of pregnancy, were evaluated. Pregnancy outcomes, including gestational age at delivery, rate of preterm delivery before 26, 28 or 30 weeks of gestation, mode of delivery (vaginal delivery or cesarean delivery), birth weight, and rate of clinical chorioamnionitis were investigated. Clinical chorioamnionitis was defined by Gibb’s criteria, with the presence of fever >37.8 ℃ as a prerequisite in addition to two other signs among maternal leukocytosis > 15,000 /mm^3^, uterine tenderness, maternal tachycardia (>160 /min), fetal tachycardia (>160 /min), and foul/purulent amniotic fluid [18].

The primary outcome measurement of this study was severe placental inflammatory pathology. Placentas were assessed by a single pathologist according to the Redline criteria [19]. Acute chorioamnionitis and funisitis were categorized as stage 0, 1, 2, or 3. For chorioamnionitis, stage 0 was defined as no inflammation of the chorioamnionic membrane. Stage 1 was defined as acute subchorionitis or acute chorioninitis and stage 2 was defined as acute chorioamnionitis. Necrotizing chorioamnionitis was defined as stage 3 acute chorioamnionitis. For acute funisitis, stage 1 was defined as umbilical phlebitis or chorionic vasculitis, stage 2 as umbilical arteritis, and stage 3 as necrotizing funisitis. Acute chorioamnionitis and funisitis were divided into two groups according to their severity. More than stage 2 chorioamnionitis or funisitis was considered severe chorioamnionitis or funisitis based on the fact that median amniotic fluid MMP-8 concentrations or WBC count were significantly increased in total stage 2 or higher chorioamnionitis in previous studies [19,20].

The secondary outcome measurements in this study were neonatal outcomes: duration of NICU stay, Apgar score at 1 and 5 minutes, necessity of mechanical ventilator support, and major neonatal morbidities including intraventricular hemorrhage (IVH, grade ≥ 3), periventricular leukomalacia (PVL), retinopathy of prematurity (ROP), respiratory distress syndrome (RDS), bronchopulmonary dysplasia (BPD), patent ductus arteriosus (PDA), necrotizing enterocolitis (NEC), sepsis (early onset, late-onset, suspected, proven), neonatal death (before 28 days), infant death (after 28 days), and neonatal composite morbidity. 

IVH was diagnosed using an imaging tool, and we limited inclusion to severe forms of IVH, where intraventricular bleeding with ventricular dilatation (grade 3) or parenchymal involvement (grade 4) was observed [21]. PVL was defined as necrosis of the white matter due to arterial ischemia in the cerebral artery watershed area, as opposed to venous congestion seen in periventricular hemorrhagic infarction [21]. ROP was diagnosed and graded by an ophthalmologist based on the International Classification of Retinopathy of Prematurity. RDS was diagnosed based on the presence of respiratory grunting and retracing, requiring oxygen supplementation with a fraction of inspired oxygen (FiO_2_) greater than 0.4, combined with ground-glass appearance and air bronchograms on chest X-ray. BPD was defined as an oxygen requirement persisting beyond 4 weeks from delivery or after a corrected age of 37 weeks [22]. PDA was defined as a symptomatic ductus arteriosus requiring pharmacological treatment or surgical ligation [23]. NEC was diagnosed and graded by a pediatrician based on modified Bell’s staging criteria, with stage 2 or 3 considered to have morbidity, which included the following criteria: presence of abdominal distension and feeding intolerance for more than 24 h with radiological evidence of intramural air, perforation, meconium plug syndrome, or definitive surgical findings of NEC [24]. Early onset neonatal sepsis was defined as a diagnosis within 7 days of birth, and late-onset was defined as a diagnosis of sepsis 1 week after birth. Proven sepsis was defined as the isolation of pathogens from body fluids, such as blood or cerebrospinal fluid. Suspected sepsis cases were determined by pediatricians based on clinical findings such as fever, hypotension, and tachycardia. Neonatal composite morbidity was defined as one or more was found among IVH ≥ grade 3, PVL, ROP ≥ grade 3, RDS, BPD, PDA, NEC ≥ stage 2b, and neonatal sepsis. 

Statistical analysis was carried out using the SPSS software (IBM SPSS Statistics for Windows, Version 27.0. IBM Corp, Armonk, NY, USA). Analysis of variance was used to compare continuous variables, whereas the chi-square test was used to compare categorical variables. Statistical significance was defined as a two-tailed *p*-value less than 0.05. For multiple comparisons, a post-hoc analysis with Bonferroni correction was applied.

Multivariable logistic regression with generalized estimating equations was used, adjusting for gestational age at delivery and stage of histologic chorioamnionitis or funisitis.

## 3. Results

A total of 939 women delivered during the study period. After excluding women who delivered due to maternal and fetal causes (*n* = 252) and who were pregnant with multifetal gestations (*n* = 257) or major fetal anomalies (*n* = 19), 411 cases were included in the analysis (Figure 1).

The patients were divided into PTL (*n* = 165), PPROM (*n* = 202) and IIOC (*n* = 44) groups according to the indication of preterm delivery. Table 1 shows maternal characteristics and pregnancy outcomes according to the indications for preterm delivery. Maternal characteristics, including maternal age, rate of multiparity, BMI, and rate of gestational diabetes, were not significantly different among the three groups. However, higher use of ART (18.2% in IIOC vs. 8.9% in PPROM vs. 6.7% in PTL, *p* = 0.036) and higher rate of chronic hypertension (4.5% in IIOC vs. 0.5% in PPROM vs. 0% in PTL, *p* = 0.006) were observed in the IIOC group than in the PTL group or PPROM group. In terms of pregnancy outcomes, the IIOC group had a significantly higher rate of women who underwent cesarean section (77.3% in IIOC vs. 52.5% in PPROM vs. 51.5% in PTL, *p* = 0.020) and higher rate of preterm delivery between 26 and 28 weeks of gestation (38.6% in IIOC vs. 23.0% in PTL vs. 20.3% in PPROM, *p* = 0.033) than the PTL group and PPROM group. The rate of clinical chorioamnionitis was higher in the PPROM group compared to the PTL group (16.8% in PPROM vs. 6.7% in PTL, *p* = 0.003), but it was not significantly different between the PPROM group and the IIOC group or between the PTL group and the IIOC group.

For placental inflammatory pathology (Table 2), histologic chorioamnionitis was observed in 64.24% of the PTL group, 76.24% in the PPROM group, and 75.00% in the IIOC group. Funisitis was observed in 41.82%, 61.88%, and 47.73% in the PTL, PPROM, and IIOC groups, respectively. The PPROM group had a significantly higher rate of severe histologic chorioamnionitis and funisitis compared to the PTL group (severe histologic chorioamnionitis, 66.30% in PPROM vs. 49.10% in PTL, *p* = 0.001; severe funisitis, 44.10% in PPROM vs. 23.60% in PTL, *p <* 0.001) (Figure 2).

On multivariable analysis, the rate of severe histologic chorioamnionitis and funisitis remained higher in the PPROM group compared to the PTL group after adjustment for gestational age at delivery and admission to delivery interval (severe histologic chorioamnionitis, OR 2.59, 95% CI 1.66–4.05; severe funisitis, OR 2.88, 95% CI 1.82–4.56) (Figure 2, Appendix A). However, there was no statistically significant difference in the rate of severe histologic chorioamnionitis and funisitis between the PPROM and IIOC group, and between the PTL and IIOC group.

For neonatal outcomes, as shown in Table 3, a higher rate of PDA was observed in the IIOC group compared to the PTL and PPROM groups (77.3% in IIOC group vs. 54.0% in PTL group vs. 54.0% in PPROM group, *p =* 0.043). The rate of PDA remained significantly higher in the IIOC group compared to PTL group with multivariable analysis after adjustment for gestational age at delivery and placental inflammatory pathology.

There were no significant differences in the rates of neonatal morbidities, neonatal or infant death, and sepsis among the PTL, PPROM, and IIOC groups. Collectively, there was no significant difference in neonatal outcomes, except for the rate of PDA between the three groups (Table 3).

## 4. Discussion

In this study, we investigated the differences in the severity of placental inflammatory pathology and neonatal outcomes according to the indication for spontaneous preterm delivery.

The principal findings of this study are as follows: First, patients with PPROM had a higher rate of severe histological chorioamnionitis and funisitis than those with PTL. Second, for neonatal outcomes, only the rate of PDA was higher in the IIOC group than in the PTL and PPROM groups. The higher rate of PDA in the IIOC group than in the PTL group remained significant after adjustment for gestational age at delivery and placental inflammatory pathology.

Placental inflammatory conditions according to the indication for preterm delivery have been reported in few studies [8,10,13,14,15]. Park et al. investigated 213 preterm gestations who delivered before 34 weeks of gestation due to PTL or PPROM and they demonstrated a higher rate of acute histologic chorioamnionitis and acute histologic chorioamnionitis with funisitis in patients with PPROM compared to those with PTL (acute histologic chorioamnionitis: 76.3% in PPROM vs. 59.2% in PTL, *p* = 0.010; acute histologic chorioamnionitis with funisitis: 50.5% in PPROM vs. 28.3% in PTL, *p* = 0.005) [15]. They described a higher rate of placental inflammation in relation to intra-amniotic infection, as a significantly higher rate of intraamniotic infection was found in women with PPROM compared to those with PTL (40.7% in PPROM vs. 16.4% in PTL, *p* = 0.001). Consistent with this previous report, our study demonstrated a higher rate of severe histological chorioamnionitis and funisitis in the PPROM group than in the PTL group. Unlike this study, we adjusted for gestational age at delivery and admission to delivery interval as histological chorioamnionitis and funisitis, which are associated with severe prematurity, and the inverse relationship between histological chorioamnionitis/funisitis and gestational age at delivery has been reported by previous studies [25,26]. Furthermore, prolongation of pregnancy decreases the risk of prematurity but may increase the risk of infection which can cause chorioamnionitis and funisitis at the same time [27,28]. In the current study, the rate of severe histologic chorioamnionitis and funisitis in women with PPROM remained significantly higher than that in women with PTL after adjustment for gestational age at delivery and admission to delivery interval (Figure 2). This can be related to the higher rate of clinical chorioamnionitis in the PPROM group than in the PTL group, as a positive correlation between clinical chorioamnionitis and placental inflammatory pathology has been reported in several previous studies [29,30].

Considering that the IIOC and PTL groups did not show significant differences in placental pathology, whereas the PPROM and PTL groups showed significant differences, it is suggested that intact membranes may affect the severity of placental inflammation. 

Similar to our study, a study comparing placental pathology among women with PTL, PPROM, and IIOC showed conflicting results from our study [31]. They reported higher grades of placental inflammatory pathology in women with IIOC compared to women with PPROM or PTL (severe histologic chorioamnionitis: 65.0% in IIOC vs. 32.0% in PPROM vs. 27.4% in PTL, *p* <0.005; severe funisitis: 35.0% in IIOC vs. 9.8% in PPROM vs. 6.8% in PTL, *p* = 0.001). This study suggested that the higher rate of placental inflammatory pathology in IIOC patients is due to the possible role of the cervical mucus plug in placental inflammation. Because the cervical mucus plug is known as a mechanical and chemical barrier to ascending infections, the IIOC group, which has fewer cervical mucus plugs, is vulnerable to ascending infection and may experience a subsequent inflammatory response. However, it is difficult to explain that the rate of severe funisitis was higher in the IIOC group compared to the PPROM group (adjusted OR 3.88, 95% CI 1.14–13.23) in this study. In the context of ascending infection, women with PPROM do not have membranes, which are definite mechanical barriers; they are more vulnerable to ascending infection than are women with IIOC or PTL. This might be due to the relatively small sample size of the previous report and the different diagnostic criteria for placental inflammation. In our study, the IIOC group demonstrated a higher rate of severe histologic chorioamnionitis and funisitis than the PTL group but a lower rate of severe histologic chorioamnionitis and funisitis than the PPROM group, although the difference was not statistically significant. This result supports a role for intact membranes in the development of placental inflammation.

There are many studies investigating the association between histologic chorioamnionitis/funisitis and neonatal outcomes and various results have been reported [32,33,34,35,36]. A meta-analysis published in 2010 investigated the association between chorioamnionitis and cerebral palsy and found that chorioamnionitis (both clinical and histological) was a risk factor for the development of cerebral palsy [36]. Similarly, many studies showed that neurologic and cognitive impairment and death were associated with severe histologic chorioamnionitis or severe funisitis [33,34,37,38]. In contrast to these reports, a decreased risk of neonatal RDS has also been reported in women with placental inflammation. According to Lee et al., among 366 preterm singleton live-born neonates, the presence of funisitis was associated with a reduced risk of developing RDS after adjusting for gestational age at delivery (OR = 0.26, 95% CI 0.14–0.50) [39]. On the other hand, another study in 2015 reported that there is no correlation between placental pathology and neonatal outcomes [17]. They investigated 339 women with PPROM who delivered before 34 weeks of gestation to identify differences in neonatal outcomes according to the stage of histologic chorioamnionitis and funisitis. They demonstrated an incremental incidence of neonatal morbidities including RDS, BPD, IVH, ROP, and composite morbidity, according to the increased stage of histologic chorioamnionitis and funisitis; however, this was not significant after adjusting for gestational age at delivery. In our study, neonatal outcomes were analyzed according to the indication for preterm delivery, but not by the severity of placental inflammation. We expected a higher rate of neonatal morbidity in the PPROM group, as this group had a higher rate of severe histologic chorioamnionitis and funisitis. However, neonatal outcomes were not statistically different among the three groups, except for PDA, which was higher in the IIOC group than in the PTL and PPROM groups. This may be related to the higher rate of preterm delivery between 26 and 28 weeks of gestation in the IIOC group. We performed multivariable analysis to determine the relationship between indication of delivery and neonatal outcomes under the control of confounding factors, including gestational age at delivery and stage of histologic chorioamnionitis and funisitis. However, there was no significant result, and the higher rate of PDA in the PTL group than in the IIOC group remained significant. Despite the significant difference in the rate of PDA in the PTL and IIOC groups, there was no significant difference in neonatal composite morbidity among three groups. This might be due to relatively small sample size in each group.

Various reports regarding neonatal morbidity and placental pathology are available; however, there is a paucity of studies that compare neonatal outcomes according to the indication for spontaneous preterm delivery. To the best of our knowledge, this is the first study to compare placental pathology and neonatal outcomes according to the indications for spontaneous preterm delivery. We compared placental pathology among pregnancies with major causes of preterm delivery (PTL, PPROM, and IIOC) and revealed different incidences of severe histologic chorioamnionitis and funisitis in each group. Neonatal outcomes did not show a positive correlation with the severity of placental pathology but differed according to the indication of spontaneous preterm delivery. Although intrauterine infection/inflammation and preterm delivery are related, the exact pathophysiology of PTL, PPROM, and IIOC remains unclear. Different extents of inflammatory changes in the placenta might be related to the presence or absence of intact membranes or other unknown factors. However, different extents of inflammatory changes in the placenta and different neonatal morbidities according to the indication for spontaneous preterm delivery can provide clues for future studies on the pathophysiology of preterm delivery.

## Figures and Tables

**Figure 1 diagnostics-12-02126-f001:**
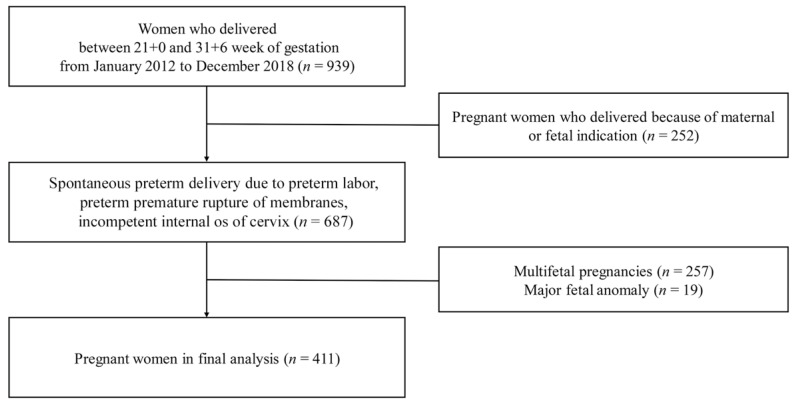
Flow chart of the enrollment of study participants.

**Figure 2 diagnostics-12-02126-f002:**
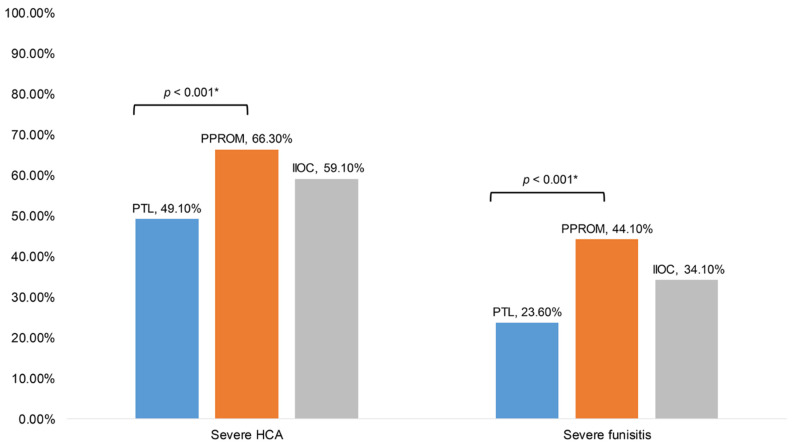
Comparison of severe histologic chorioamnionitis (HCA) or funisitis among women who delivered due to preterm labor (PTL), preterm premature rupture of membranes (PPROM), or incompetent internal os of the cervix (IIOC). * Significant after adjustment for gestational age at delivery and admission to delivery interval.

**Table 1 diagnostics-12-02126-t001:** Maternal characteristics and pregnancy outcomes according to the causes of preterm delivery.

	PTL (*n* = 165)	PPROM (*n* = 202)	IIOC (*n* = 44)	*p*-Value			
				PTL vs. PPROM *	PPROM vs. IIOC *	PTL vs. IIOC *	PTL vs. PPROM vs. IIOC
Age (year, mean ± SD)	33.30 ± 4.02	33.05 ± 3.91	33.09 ± 4.03	0.552	0.950	0.763	0.833
Multiparity	80/165 (48.5%)	91/202 (45.0%)	17/44 (38.6%)	0.512	0.437	0.244	0.244
Use of ART	11/165 (6.7%)	18/202 (8.9%)	8/44 (18.2%)	0.428	0.070	0.018	0.036
BMI (kg/m^2^)	24.62 ± 3.96	24.41 ± 4.20	25.72 ± 4.14	0.658	0.086	0.138	0.213
Use of ACS	149/165 (90.3%)	183/202 (90.6%)	42/44 (95.5%)	0.925	0.296	0.279	0.410
GDM	14/165 (8.5%)	22/202 (10.9%)	3/44 (6.8%)	0.441	0.418	0.719	0.600
Chronic hypertension	0/165 (0%)	1/202 (0.5%)	2/44 (4.5%)	0.365	0.027	0.006	0.006
Cesarean delivery	85/165 (51.5%)	106/202 (52.5%)	34/44 (77.3%)	0.193	0.003	0.002	0.020
GAD (days, mean ± SD)	191.17 ± 20.33	194.44 ± 19.49	189.43 ± 14.82	0.118	0.060	0.527	0.144
PTD at <26 weeks	55/165 (33.3%)	60/202 (29.7%)	15/44 (34.1%)	0.456	0.567	0.925	0.778
PTD at 26 + 0~27 + 6 weeks	38/165 (23.0%)	41/202 (20.3%)	17/44 (38.6%)	0.526	0.009	0.037	0.033
PTD at 28 + 0~29 + 6 weeks	35/165 (21.2%)	38/202 (18.8%)	7/44 (15.9%)	0.567	0.652	0.435	0.394
PTD at 30 + 0~31 + 6 weeks	37/165 (22.4%)	63/202 (31.2%)	5/44 (11.4%)	0.061	0.008	0.104	0.012
Admission to delivery interval (days)	2.00 [0–58]	5.00 [0–86]	2.50 [0–46]	<0.001	0.229	0.350	0.001
Clinical chorioamnionitis	11/165 (6.7%)	34/202 (16.8%)	2/44 (10.7%)	0.003	0.037	0.605	0.249
Severe HCA	81/165 (49.1%)	134/202 (66.3%)	26/44 (59.1%)	0.001	0.361	0.238	0.014
Severe funisitis	39/165 (23.6%)	89/202 (44.1%)	15/44 (34.1%)	<0.001	0.225	0.159	0.004

PTL; preterm labor, PPROM: preterm premature rupture of membranes, IIOC: incompetent internal os of the cervix, SD; standard deviation, ART: assisted reproductive technologies, BMI; body mass index, ACS: antenatal corticosteroids, GDM; gestational diabetes mellitus, GAD; gestational age at delivery, PTD; preterm delivery. * A value of *p* < 0.016 was regarded as statistically significant as Bonferroni correction for multiple comparisons was applied.

**Table 2 diagnostics-12-02126-t002:** Placental inflammatory pathology stage according to the causes of preterm delivery.

	PTL (*n* = 165)	PPROM (*n* = 202)	IIOC (*n* = 44)	*p*-Value		
				PTL vs. PPROM	PPROM vs. IIOC	PTL vs. IIOC
**Histologic chorioamnionitis stage**						
0	59/165 (35.76%)	48/202 (23.76%)	11/44 (25.00%)	0.010	0.633	0.482
1	25/165 (15.15%)	20/202 (9.90%)	7/44 (15.91%)			
2	25/165 (15.15%)	45/202 (22.28%)	10/44 (22.73%)			
3	56/165 (33.94%)	89/202 (44.06%)	16/44 (36.36%)			
**Funisitis stage**						
0	96/165 (58.18%)	77/202 (38.12%)	23/44 (52.27%)	< 0.001	0.343	0.195
1	30/165 (18.18%)	36/202 (17.82%)	6/44 (13.64%)			
2	26/165 (15.76%)	71/202 (35.15%)	13/44 (29.55%)			
3	13/165 (7.88%)	18/202 (8.91%)	2/44 (4.54%)			

PTL; preterm labor, PPROM: preterm premature rupture of membranes, IIOC: incompetent internal os of the cervix.

**Table 3 diagnostics-12-02126-t003:** Neonatal outcome according to the causes of preterm delivery.

	PTL (*n* = 165)	PPROM (*n* = 202)	IIOC (*n* = 44)	*p*-Value			
				PTL vs. PPROM *	PPROM vs. IIOC *	PTL vs. IIOC *	PTL vs. PPROM vs. IIOC
Birth weight (kg, mean ± SD)	1.09 ± 0.42	1.16 ± 0.44	1.05 ± 0.35	0.147	0.074	0.475	0.165
1 min Apgar score < 4	33/165 (20.0%)	33/165 (22.0%)	7/44 (15.9%)	0.193	0.859	0.540	0.277
5 min Apgar score < 7	39/165 (23.6%)	38/202 (18.8%)	9/44 (20.5%)	0.259	0.802	0.656	0.383
Duration of NICU stay (days, mean ± SD )	75.66 ± 56.72	72.17 ± 54.73	82.80 ± 46.28	0.550	0.232	0.443	0.486
Ventilator treatment	150/165 (90.9%)	167/202 (82.7%)	42/44 (95.5%)	0.022	0.032	0.327	0.598
Neonatal death	19/165 (11.5%)	16/202 (7.9%)	3/44 (6.8%)	0.244	0.804	0.367	0.208
Infant death	11/146 (7.5%)	19/186 (10.2%)	1/41 (2.4%)	0.398	0.112	0.239	0.713
Neonatal morbidity							
IVH (≥grade 3)	14/154 (9.1%)	25/198 (12.6%)	6/43 (4.9%)	0.294	0.814	0.351	0.257
PVL	14/154 (9.1%)	12/198 (6.1%)	2/43 (7.1%)	0.281	0.720	0.346	0.212
ROP (≥grade 3)	25/143 (17.5%)	31/178 (17.4%)	8/41 (19.5%)	0.987	0.752	0.765	0.828
RDS	139/165 (84.2%)	152/202 (75.2%)	40/44 (90.9%)	0.034	0.023	0.262	0.766
BPD	37/140 (26.4%)	43/179 (24.0%)	8/41 (9.1%)	0.623	0.537	0.368	0.368
PDA	87/161 (54.0%)	108/200 (54.0%)	34/44 (77.3%)	0.994	0.005	0.005 **	0.043
NEC (≥grade 2)	11/165 (6.7%)	17/202 (8.4%)	1/44 (2.3%)	0.530	0.156	0.266	0.665
Neonatal composite morbidity ^a^	106/165 (64.2%)	129/202 (63.9%)	36/44 (81.8%)	0.940	0.022	0.026	0.117
Sepsis	25/163 (15.3%)	35/201 (17.4%)	9/44 (20.5%)	0.596	0.634	0.416	0.402
Early onset neonatal sepsis	18/163 (11.0%)	22/201 (10.9%)	5/44 (11.4%)	0.976	0.936	0.952	0.976
Late onset neonatal sepsis	11/163 (6.7%)	18/201 (9.0%)	4/44 (9.1%)	0.439	0.977	0.595	0.464
Suspected sepsis	9/163 (5.5%)	13/201 (6.5%)	3/44 (6.8%)	0.706	0.932	0.744	0.682
Proven sepsis	21/163 (12.9%)	26/201 (12.9%)	7/44 (15.9%)	0.988	0.601	0.603	0.694

PTL; preterm labor, PPROM: preterm premature rupture of membranes, IIOC: incompetent internal os of the cervix, SD; standard deviation, NICU; neonatal intensive care unit, IVH; intraventricular hemorrhage, PVL; periventricular leukomalacia, ROP; retinopathy of prematurity, RDS; respiratory distress syndrome, BPD; bronchopulmonary dysplasia, PDA; patent ductus arteriosus, NEC; necrotizing enterocolitis. ^a^ Defined as having more than 1 of the following: IVH (grade 3 or higher), PVL, ROP (grade 3 or higher), RDS, BPD, PDA, NEC (grade 2 or higher), sepsis. * A value of *p* < 0.016 was regarded as statistically significant as Bonferroni correction for multiple comparisons was applied. ** Significant after adjustment for gestational age at delivery and histologic chorioamnonitis or funisitis.

## Data Availability

The data presented in this study are available on request from the corresponding author.

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
