# Peer review of "Do the Causes of Spontaneous Preterm Delivery Affect Placental Inflammatory Pathology and Neonatal Outcomes?"

_diagnostics, 2022, doi:10.3390/diagnostics12092126_

Round 1

Reviewer 1 Report

in this manuscript authors investigated the severity of histologic chorioamnionitis /funisitis according to the indication for preterm delivery and their corresponding neonatal outcomes. They found that PPROM showed a higher rate of severe histological chorioamnionitis/funisitis compared to PTL. Moreover, authors observed a higher rate of PDA  in IIOC compared to PTL and PPROM. 

the manuscript is clear and generally well written but the introduction needs to be implemented in order to highlight the distinction between PTL and PPROM. In particular: 

Introduction: Authors must give a definition of PTL, PPROM and IIOC. Moreover, It deserves to be pointed out that PPROM is mainly due to the inflammation (induced by the infection) which cytokines weaken the foetal membranes (as previously demonstrated PMID: 26739007). The same mechanism also contributes to funisitis onset by altering the placental barrier (PMID: 24768095). These can explain the different clinical manifestations between PTL and PROM (PMID: 27090052) and further highlight the results found by the authors ( lines 242-243)

Line 333PPPOM?

Author Response

We would like to deeply appreciate your valuable comments and the constructive suggestions which will be helpful to further improve the manuscript. Following are point-by-point response to your comments and we hope that we successfully addressed each point raised.

Comment #1:

in this manuscript authors investigated the severity of histologic chorioamnionitis /funisitis according to the indication for preterm delivery and their corresponding neonatal outcomes. They found that PPROM showed a higher rate of severe histological chorioamnionitis/funisitis compared to PTL. Moreover, authors observed a higher rate of PDA in IIOC compared to PTL and PPROM. 

the manuscript is clear and generally well written but the introduction needs to be implemented in order to highlight the distinction between PTL and PPROM. In particular: 

Introduction: Authors must give a definition of PTL, PPROM and IIOC. Moreover, It deserves to be pointed out that PPROM is mainly due to the inflammation (induced by the infection) which cytokines weaken the foetal membranes (as previously demonstrated PMID: 26739007). The same mechanism also contributes to funisitis onset by altering the placental barrier (PMID: 24768095). These can explain the different clinical manifestations between PTL and PROM (PMID: 27090052) and further highlight the results found by the authors (lines 242-243)

Response:

We appreciate your thoughtful comments and the constructive suggestions. We specified the definition of PTL, PPROM and IIOC in the introduction section. Also, we added the possible pathophysiology of PPROM as following and highlighted the changes in the manuscript.

“PTL is the most common cause of spontaneous preterm delivery and is defined as regular uterine contractions with cervical change before 37 weeks of gestation. PPROM is defined as membrane rupture before 37 weeks of gestation and causes 30~35% of preterm delivery. IIOC is painless cervical dilatation without PTL or PPROM and is also an important cause of preterm delivery.” (Line 32~36)

“It is reported that particular cytokines which can weaken the placental barrier and fetal membranes are released when chorioamnionitis or funisitis develops, and this might cause premature rupture of membranes.(Line 38~40)

Comment #2: Line 333: PPPOM?

Thank you for your comment. We replaced ‘PPPOM’ in line 333 with ‘PPROM’ and highlighted the changes in the manuscript.

Reviewer 2 Report

Jang et al investigated the severity of histologic chorioamnionitis and funisitis according to the indication for preterm delivery and corresponding neonatal outcomes. They compared placental inflammatory pathologic lesions and neonatal outcomes in patients who delivered between 21 and 31st week of gestation with preterm labor (PTL), preterm rupture of membranes (PPROM) and incompetent cervix (IIOC). The authors showed PPROM patients had significantly higher rates of severe histologic chorioamnionitis and funisitis compared to PTL. The incidence of severe inflammation was not different between PPROM and IIOC and PTL and IIOC groups. Amongst the neonatal outcomes patent ductus arteriosus was found to be significantly more seen in IIOC compared to PTL and PPROM groups. This is a nicely written and well designed study. I would like to congratulate the authors.

After my review, I would like to share my comments and suggestions.

1-Please open “PDA” as patent ductus arteriosus in abstract.

2-Introduction is brief and informative but can you please clarify better what you want to mean in the last sentence of the second paragraph of introduction, lines 55 to 57, “Although there might be……. On the cause of preterm delivery.”.

3-I think authors acted smart by including patients before 32 weeks since most of these preterm deliveries can be associated with inflammatory process.

4-If authors would like to emphasize the significant results in their tables (p values etc), I suggest that these areas are bolded instead of red color because depending on the reader’s access type, the reader may not realize different color if it is not a colored print.

5-Especially in patients with PPROM and IIOC, time interval between diagnosis and delivery is very important regarding the risk for infection/inflammation. Clinical recommendation and trend is to keep these patients pregnant for a while after the diagnosis. I am curious about this data since PPROM patients had higher rates of severe histologic chorioamnionitis. I wonder if their result could have been affected by the time interval until delivery. Do the authors have this information and can they include this in the manuscript? If not, I think it is important that this is included in the discussion section and used as a limitation of this study.

6-Nonsignifcant neonatal outcomes can be due to the low incidence in small groups. Although composite outcome, (which is commonly used because of this reason) is not different between groups in this study, this should be included as a limitation of this study.

7- There is good data showing increased risk of intraamniotic infection/inflammation and placental inflammation as high as 50-80% in patients with cervical incompetence (Romero R AJOG 1992 and Lee SE, Romero R, AJOG 2008 are some of them). Therefore, it is not too surprising to see IIOC group had higher rates of histologic chorioamnionitis and funisitis compared to PTL (regards to lines 288-292).

Thank you,

Author Response

We would like to deeply appreciate your valuable comments and the constructive suggestions which will be helpful to further improve the manuscript. Following are point-by-point response to your comments and we hope that we successfully addressed each point raised.

Comment #1:

Jang et al investigated the severity of histologic chorioamnionitis and funisitis according to the indication for preterm delivery and corresponding neonatal outcomes. They compared placental inflammatory pathologic lesions and neonatal outcomes in patients who delivered between 21 and 31st week of gestation with preterm labor (PTL), preterm rupture of membranes (PPROM) and incompetent cervix (IIOC). The authors showed PPROM patients had significantly higher rates of severe histologic chorioamnionitis and funisitis compared to PTL. The incidence of severe inflammation was not different between PPROM and IIOC and PTL and IIOC groups. Amongst the neonatal outcomes patent ductus arteriosus was found to be significantly more seen in IIOC compared to PTL and PPROM groups. This is a nicely written and well designed study. I would like to congratulate the authors.

After my review, I would like to share my comments and suggestions.

Please open “PDA” as patent ductus arteriosus in abstract.

Response:

Thank you for your comment. We replaced ‘PDA’ in abstract with ‘patent ductus arteriosus’ and highlighted the changes in the manuscript.

Comment #2:

Introduction is brief and informative but can you please clarify better what you want to mean in the last sentence of the second paragraph of introduction, lines 55 to 57, “Although there might be……. On the cause of preterm delivery.”

Response:

We appreciate your valuable comment. We intended to emphasize the paucity of studies investigating the relationship between placental pathology and indication of preterm delivery and additionally, association between placental pathology and indication of preterm delivery. However, we admit that the sentence is confusing and we changed the text as following and highlighted in the manuscript.

“Although several studies have been conducted on the relationship between the severity of placental pathology and neonatal prognosis in preterm mothers, there are few studies on how the severity of placental pathology differs depending on the cause of preterm delivery.”(Line62~64)

Comment #3:

I think authors acted smart by including patients before 32 weeks since most of these preterm deliveries can be associated with inflammatory process.

Response:

We sincerely appreciate your positive and encouraging comments.

Comment #4:

If authors would like to emphasize the significant results in their tables (p values etc), I suggest that these areas are bolded instead of red color because depending on the reader’s access type, the reader may not realize different color if it is not a colored print.

Response:

Thank you for your valuable comment. We replaced red color with bolded letters.

Comment #5:

Especially in patients with PPROM and IIOC, time interval between diagnosis and delivery is very important regarding the risk for infection/inflammation. Clinical recommendation and trend is to keep these patients pregnant for a while after the diagnosis. I am curious about this data since PPROM patients had higher rates of severe histologic chorioamnionitis. I wonder if their result could have been affected by the time interval until delivery. Do the authors have this information and can they include this in the manuscript? If not, I think it is important that this is included in the discussion section and used as a limitation of this study.

Response:

Thank you for your constructive suggestions and thoughtful comment. We added data of admission to delivery interval (days) in Table 1 and highlighted. We investigated severe histologic chorioamnionitis and funisitis using multivariable logistic regression, adjusting for gestational age at delivery and admission to delivery interval. In our study, the rate of severe histologic chorioamnionitis and funisitis remained higher in PPROM group compared to PTL group after adjustment for gestational age at delivery and admission to delivery interval (severe histologic chorioamnionitis, OR 2.594, 95% CI 1.660-4.054; severe funisitis, OR 2.883, 95% CI 1.824-4.559). However, there was no statistically significant difference between PPROM and IIOC group (severe histologic chorioamnionitis, OR 0.554, 95% CI 0.281-1.090; severe funisitis, OR 0.601, 95% CI 0.306-1.180), and between PTL and IIOC group (severe histologic chorioamnionitis, OR 1.368, 95% CI 0.696-2.692; severe funisitis, OR 1.587, 95% CI 0.787-3.201). We added the result of analysis in figure 2 (footnote) and described in the manuscript as follows. Thank you again for your comments.

“On multivariable analysis, the rate of severe histologic chorioamnionitis and funisitis remained higher in PPROM group compared to PTL group after adjustment for gestation-al age at delivery and admission to delivery interval (severe histologic chorioamnionitis, OR 2.594, 95% CI 1.660-4.054; severe funisitis, OR 2.883, 95% CI 1.824-4.559) (Figure 2). However, there was no statistically significant difference in the rate of severe histologic chorioamnionitis and funisitis between PPROM and IIOC group, and between PTL and IIOC group.” (Line 225~230)

Comment #6:

Nonsignifcant neonatal outcomes can be due to the low incidence in small groups. Although composite outcome, (which is commonly used because of this reason) is not different between groups in this study, this should be included as a limitation of this study.

Response:

We agree with the reviewer’s opinion. Although the total number of enrolled patients are not small, the number of patients in each group (PTL, PPROM, IIOC) is not large. We added about this limitation in the discussion section line 335-338 as following and highlighted in the manuscript.

Despite the significant difference in the rate of PDA of PTL and IIOC group, there was no significant difference in neonatal composite morbidity among three groups. This might be due to relatively not large sample size in each group.”(Line 335-338)

Comment #7:

There is good data showing increased risk of intraamniotic infection/inflammation and placental inflammation as high as 50-80% in patients with cervical incompetence (Romero R AJOG 1992 and Lee SE, Romero R, AJOG 2008 are some of them). Therefore, it is not too surprising to see IIOC group had higher rates of histologic chorioamnionitis and funisitis compared to PTL (regards to lines 288-292).

Response:

Thank you for your comment and we agree to the reviewer's opinion.

Reviewer 3 Report

Dear Authors I admire the great amount

of work that you done however

I need to put up to few things. 

The results of your assessment are obvious. 

1. The mucous plug and intact membranes are

better protectors for growing fetus. It was

proven in many researches. These kind of

conclusions aren’t acceptable. 

2. In the Inclusion criteria you write that the 

gestational age considered for the study was 21-31 weeks  

However in the Statistical Table you divide your

patients : < 26, 26-27, 27-28 weeks. Where is the rest of patients?

3. The main reason for rejecting this article is

that you can’t compare neonatal results like RDS or others

only because of the indication for PTD but you need 

to compare them to gestational age, eg IVH in 22 weeks 

is more common than in 30 weeks  

If you will rewrite the article you could try to resubmit

I suggest a different comparison and statistical 

inquisitiveness

Author Response

We would like to deeply appreciate your valuable comments and the constructive suggestions which will be helpful to further improve the manuscript. Following are point-by-point response to your comments and we hope that we successfully addressed each point raised.

Comment #1:

Dear Authors

I admire the great amount of work that you done however I need to put up to few things.

The results of your assessment are obvious.

The mucous plug and intact membranes are better protectors for growing fetus. It was proven in many researches. These kind of conclusions aren’t acceptable.

Response:

We appreciate your comment and we agree that the cervical mucus plug and intact membranes are acting as a barrier from ascending infection/inflammation. As our data revealed higher incidence of severe histologic chorioamnionitis and funisitis in PPROM group compared to PTL group, we suggested the protective effect of cervical mucus plug and intact membranes. We hope this could help clarifying the authors’ intention.

Comment #2:

In the Inclusion criteria you write that the gestational age considered for the study was 21-31 weeks

However in the Statistical Table you divide your patients : < 26, 26-27, 27-28 weeks. Where is the rest of patients?

Response:

Thank you for your comment. We added data of patients who delivered at 30+0 ~ 31+6 weeks of gestation in Table 1 and highlighted.

Comment #3:

The main reason for rejecting this article is that you can’t compare neonatal results like RDS or others only because of the indication for PTD but you need to compare them to gestational age, eg IVH in 22 weeks is more common than in 30 weeks

If you will rewrite the article you could try to resubmit

I suggest a different comparison and statistical inquisitiveness

Response:

Thank you for your valuable comments. We performed multivariable analysis to investigate the relationship between neonatal outcomes and indication of preterm delivery adjusting for gestational age at delivery and histologic chorioamnionitis or funisitis but there was no significant findings. We have demonstrated this results in Table 3.